# Factors of Trust in Immediate Leaders: An Empirical Study in Police Service Environment

**DOI:** 10.3390/ijerph16142525

**Published:** 2019-07-15

**Authors:** Nina Tomaževič, Aleksander Aristovnik

**Affiliations:** Faculty of Public Administration, University of Ljubljana, 1000 Ljubljana, Slovenia

**Keywords:** trust, job satisfaction, leadership, police, healthy work environment

## Abstract

The study has two objectives—first, to examine the dimensionality of police service employees’ job satisfaction and their assessment of the enablers from the Common Assessment Framework, analyzing these facets at different organizational levels, and second, to identify the impact of selected facets of job satisfaction on trust in one’s immediate leader. The study is based on a comprehensive on-line questionnaire where 1209 responses from police service employees were acquired, and for the data analysis, factor analysis was first used to formulate the factors of job satisfaction facets and of the Common Assessment Framework enablers. Second, structural equation modelling was performed to identify the correlations of the studied variables. The Common Assessment Framework enabler Strategy/Leadership was significantly correlated with the enabler Processes and had an indirect impact on Trust through direct impacts on Leadership style in the organizational unit and Autonomy/Tasks. Both of the latter determinants had a significant influence on Trust in one’s immediate leader. The confirmed the impact of determinants from different hierarchical levels indicates the importance of efforts by managers at all organizational levels should the police management intend to achieve and maintain high levels of trust in one’s immediate leader in police service work environment.

## 1. Introduction

Any relationship and therefore any business, in either the private or public sector, involves two or more different stakeholders trying to establish connections and cooperation both outside and within their operations. Any type of cooperation requires a specific level of mutual trust for stakeholders to feel psychologically safe. This is especially important within businesses—among co-workers themselves, and even more so, among employees and their leaders. Building trust within an organization is usually a long-term process, but when achieved, it brings numerous advantages and makes cooperation easier and more efficient. In the work environment, a healthy level of trust among co-workers and between representatives of different hierarchical levels improves the quality of working life, and consequently, job satisfaction, employee engagement, commitment, and many other constructs from the Human Resources Management field. Once trust is established, it has to be maintained.

Many practitioners and scholars, especially since the late 1950s, have looked for the consequences a healthy level of trust brings to intra-organization relationships. They found that trust influences different aspect of business, such as managerial and organizational effectiveness [1,2,3], team performance [4,5], job satisfaction [3,6,7], the commitment-withdrawal relationship [8,9], organizational citizenship behavior [10,11], problem-solving [12], innovative behavior [10], cooperation [9,13], employee involvement [3], creativity [5], and many others.

On the other hand, there are factors/determinants that influence trust like authentic leadership [10], CEO (Chief Executive Officer) relational leadership [14], leader action and practices (transformational leadership, transactional leadership, perceived organizational support [15], participative decision-making, unmet expectations, justice etc.), follower attributes (propensity to trust), relationship attributes (length of relationship) [16], managerial trustworthy behavior [17] and the quality and quantity of information [3]. In their qualitative meta-analysis of trust in supervisor-subordinate relationships in 2015, Nienaber, Romeike, Searle and Schewe [11] discovered four distinct clusters of trust antecedents: Supervisor attributes, subordinate attributes, interpersonal processes, and organizational characteristics.

In recent years, research on trust and leadership in police settings has grown in both quantity and quality. Trust has been examined as a phenomenon between other stakeholders and police service employees (e.g., the public) or a phenomenon that exists within the police (among colleagues and towards superiors). Leadership has mainly been studied by analyzing managers’ competencies, characteristics, roles, behaviors and leadership styles. Many studies of police leadership have focused on a lower level of officers, but only few studies examined the macro-levels of police organizations [18]. Therefore, the presented study extended the focus to different organizational levels. The main aim of this analysis was to determine the strength of the influence of different determinants from three organizational levels on police service employees’ trust in their superiors.

The paper is structured as follows. In the next section, the literature on trust, organizational levels, and selected determinants of trust is reviewed, followed by a description of the context of the police service in Slovenia. Section 3 describes the methodology (participants, instruments, data analysis), Section 4 presents the results, while the last section consists of a discussion and conclusion.

## 2. Literature Review and Hypothesized Structural Relationship Model

### 2.1. Trust in Leaders

There are many definitions of trust. From its psychological structure, trust is an emotion—it has a direction, strength, and duration. According to its value, it has two poles: Trust and distrust. It determines the relational component towards things, phenomena, other people, and oneself. The word originates from an anthropologic category of believing and encompasses five important parts: conviction, entrustment, hope, trust, and self-trust [19]. Trust can also be defined as part of a relationship between two people and involves the voluntary acceptance by the trustor of risk based on the actions of the other party [3]. Mayer, Davis, and Schoorman [20] define trust as ‘the willingness of a party to be vulnerable to the actions of another party’. Feltman [21] similarly defines trust as ‘choosing to make something important to you vulnerable to the actions of someone else’. Ferdowsian [22] defines trust as a quality or virtue, which means confidence, certainty, and reliance. Jones and George [23] see trust as an expression of confidence by a party that their vulnerability will not be exploited and that they will not be harmed by the behaviors or actions of the other party.

Trust is a core relational construct, commonly conceptualized as a psychological state in which individuals make themselves vulnerable in a relationship based on expectations, assumptions, or beliefs that another’s future behaviors will be positive, beneficial, or favorable [14,24]. Followers willingly make themselves vulnerable to leaders whom they have previously experienced as competent, honest, and willing to genuinely look out for them [16]. Earlier studies on relationships between employees and leaders showed that people are initially trusting of other people [25], that the level of trust at the point of betrayal influences subsequent reactions and the possibilities of its repair [26,27], and that trust develops over time [28].

Trust has gained wide acceptance in the literature as a means for improving individual, group and organizational performance [3]. According to Ferdowsian [22], trust brings many positive effects such as lower stress, shorter time-to-decision, fewer conflicts, and stronger cohesion. While trust should not be seen as the ultimate solution to all organizational problems, teams may become extremely unproductive if the individual members feel tense, unsatisfied, and less emotionally committed [29]. In the presented study, the trust was examined by asking the respondents (1) whether they trust their immediate superior and (2) whether they receive all information needed for execution of their job.

### 2.2. Determinants of Trust at the Macro-Organizational Level

On the macro-organizational level, the elements of the business excellence model for the public sector were selected as the key determinants of trust. When effectively applied, it can improve the performance of public organizational level, many elements must be taken into account when looking for the determinants of trust. Some of them are definitely those touching on strategic management, leadership at the top management level, and the processes designed in order for operations to run smoothly. In our study, the macro-organizational level was represented by the General Police Directorate, which is in charge and authorized for the strategic elements mentioned above. The determinants of trust at the macro-organizational level were designed on the basis of the Common Assessment Framework (CAF) model and the respondents had to assess their level of satisfaction with a specific CAF criterion. The CAF is a self-evaluation model that can help establish a quality/excellence philosophy [30]. It is a costless tool for public sector organizations’ self-assessment, allowing them to achieve specific improvement measures and bring change and growth in the excellence of public services [30,31,32]. On the other hand, already several studies point to varying degrees of failure in implementing excellence models [33,34,35] because business excellence in the public sector faces the following barriers [36]: Lack of commitment from top management, weaknesses in training and HRM (Human Resources Management), weaknesses in the management information systems, and a non-conducive organizational culture.

The model consists of nine criteria (five enablers and four results), with each being divided into sub-criteria (28 in total). These are subdivided into elements/questions (212 in total). The enablers that should assure excellence (results) are: (1) Leadership, (2) People, (3) Strategy and Planning, (4) Partnership and Resources, and (5) Processes [37]. Enablers 2 to 5 are determined by the management and are brought to employees by executing leadership approaches. In order to make this execution as effective and efficient as possible and to achieve excellent results, it is thus very important how employees perceive these enablers and the managers’ attitude to them. According to Bou, Escrig, Roca and Beltrán [38] and Calvo, Picón, Ruiz and Cauzo [39], the EFQM (European Foundation for Quality Management) criteria (and therefore of CAF) have a complex structure where changes in one element may be related to changes in other elements, thus implying interdependence among the criteria. In the study, presented in the paper, the main focus was on the correlation between the CAF enablers themselves and their impact on facets of job satisfaction (job satisfaction was not measured in general, but according to the selected elements, e.g., satisfaction with (1) relationships among the staff, (2) possibility of participating in decision-making and organization, (3) style of leading the organizational unit, (4) possibility of realizing one’s abilities, (5) possibility of performing work autonomously, (6) supervision of work, (7) feeling of belonging to the staff, (8) reward system, (9) salary, (10) promotion system, (11) payment of overtime, (12) professional training system, (13) public attitude to the police, (14) psycho-hygienic care for police officers, (15) functioning of the police trade union, (16) security of employment, (17) volume of tasks, (18) administrative tasks, (19) volume of regulations, work guidelines, (20) working conditions (equipment, premises), (21) job location, (22) variety of tasks, (23) work with people, and (24) working hours), which is one of many results criteria of the CAF model.

### 2.3. Determinants of Trust at the Mezzo-Organizational Level

On the mezzo-organizational level, there are elements that influence trust from the level of middle management. At this level, the leadership style in an organizational unit (in our case police station) was found as an important factor of trust [5]. According to Grover et al. [26], trust is essential for leadership. It plays a significant role in the relationship between leaders and subordinates [40,41,42]. Leadership is one of those obvious elements required for integrity and accountability in policing, but how to generate and maintain professional leadership is a difficult question [43]. Leaders are those who set direction and motivate, inspire and align people accordingly [44]. They are a source of both guidance and inspiration. Effective leaders have the ability to inspire followers to do work that is well beyond the minimum required of them [45]. Leadership is considered a set of positive traits, with those lacking the requisite skills not being true leaders. However, extant literature tends to regard persons in positions of power who abuse their authority, commit acts of corruption, engage in untoward conduct, or fail to shoulder the mantle of their responsibilities as not being actual leaders despite them holding a leadership position [46]. According to Balley [47] and Kellerman [48], the ‘leadership-is-a-positive-action’ orientation overlooked the reality that even effective and well-regarded leaders are not perfect; they have shortcomings, sometimes use less-than-ideal means in pursuing the desired ends. Since the late 1980s, a growing body of leadership scholarship has considered leadership as more than just a positive process and outcome [46].

Interest in police leadership research largely developed in North America in the—in response to the civil rights movement and the social unrest in the previous decade and coinciding with the early development of community policing [49]. The elements of police leadership are also a vital concern for officers in the field. Police managers have been exhorted to recognize that leadership can be exercised at any level of the department [45]. According to Baker [50], effective leadership is exercised by police managers in different ways, depending on their rank in the department. Senior management should be in charge of developing and sharing the vision, charting the journey by establishing strategic objectives, and practicing collaboration and task delegation. Police middle managers coordinate and plan, mentor and coach, build teams and empower and reward their subordinates. First-line supervisors provide leadership by example, supervising and training teams, while evaluating their performance.

According to Campbell and Kodz [49], leadership is a complex research area and across all sectors there is ambiguity over which styles and behaviors are the most effective. There is virtually no reliable evidence on the impacts of police leadership styles and behaviors make on the ground. Leadership is an evolving process, and across all sectors, there is little certainty concerning which styles and behaviors produce the most effective outcomes. However, in the presented study the predecessors and antecedents of satisfaction with the leadership of an organizational unit were analyzed, as assessed by the police service employees.

### 2.4. Determinants of Trust at the Micro-Organizational Level

At the level of an individual job (micro-organizational level), task autonomy is one of the most important job characteristics. It can be defined as the degree to which employees have freedom and discretion, the right of self-government, independence, and accountability to carry out their operationally defined tasks and schedule their work [51,52,53]. It makes individuals’ work more meaningful through jobs that consist of a broad scope of tasks and that enable individuals to see the significance of their contributions [54]. The authority to make independent decisions that come with autonomy, conveys trust in employees’ abilities, and generates a sense of responsibility to the organization [55]. Troyer, Mueller and Osinsky [56] claim that a high level of job autonomy allows employees to decide how to perform their work.

When studying the consequences of job autonomy, Hornung and Rousseau [57] found that autonomy on the job is perhaps the primary work characteristic in shaping worker attitudes, motivation and behavior. Promoting worker autonomy itself can be a critical precursor to the successful implementation of certain forms of organizational change. Work autonomy directly contributes to employees’ job satisfaction [58,59,60]. Autonomy has also been found to be an important predictor of proactive outcomes, including suggesting improvements [61]. Sia and Appu [62] discovered that the three dimensions of work autonomy, namely, work method autonomy, work schedule autonomy and work criteria autonomy, have a direct positive contribution to workplace creativity. The same has been found regarding task complexity. According to Yan, Chong and Mak [63], the execution of task autonomy significantly and positively relates to performance.

Anand, Chhajed and Delfin [64] claim that autonomy and trust in leadership are the two areas of employee satisfaction that directly result from organizational support and can be impacted by top management’s actions. The study, presented in the paper focused on the impact of police service employee satisfaction with top management’s strategies, leadership approaches, and processes on satisfaction with the autonomy of tasks and its influence on trust.

### 2.5. Hypothesized Structural Relationship Model

While there is an abundance of findings, research on the topic of trust is fragmented. Many studies include trust in their model, but do not treat it as the core variable of interest. The main purpose of our study is to contribute to understanding of the impact of strategies, leadership and processes as determinants at the macro-organizational level (as established in the Common Assessment Framework—CAF) on police service employees’ trust in leaders, with (1) leadership style in organizational unit (LSOU) on the mezzo-organizational level; and (2) autonomy and tasks on the micro-organizational (‘individual’) level as two mediating variables. While previous studies provided empirical support for many specific links that are included in our hypothesized model (see Figure 1), none of the previous studies attempted to integrate these three-level organizational structure variables into a comprehensive model. Using structural equation modelling (SEM), the hypothesized research model shown in Figure 1 is tested.

## 3. Materials and Methods

### 3.1. Participants and Instruments

The study was designed as an explanatory quantitative research applying the online questionnaire. The later was self-designed as a combination of (1) a survey conducted in police 2 years before our survey and (2) claims, adapted for police service on the basis of the CAF model criteria. There were 1209 respondents included in the survey sample (response rate with fully answered questionnaires was 13.7%), all employed by the Slovenian police service, with 87.7% of them being male. The respondents had worked for the police for 17.3 years on average. Secondary school or a lower educational level was completed by 44% of the respondents, while 56% of them achieved a higher or university education. A leadership position in police units was held by 16.7% of the respondents. While 47.4% of the respondents were doing fieldwork, 52.6% worked in the office. Their work was mainly (28.5%) in the field of the detection and investigation of criminal offences—the other fields of work were ensuring public order and piece, road safety, security of state borders, special areas (forensics, security of persons and facilities, back-office activities (training, information technology, analytics, control, etc.), criminal intelligence activities, and ‘other’.

In order to ensure a high number of responses, we had asked the police management and the trade union leaders to promote the survey among their colleagues. The first part of the questionnaire contained the following variables:

- demographic questions: Gender, age, educational level (secondary school and less, college, higher and university education, postgraduate), length of service in the police (in years); and

- job-related questions, like job location (field or office workers), job position (police officer, senior police officer, leader of an organizational unit), distance from home to the workplace (in km).

The second part of the questionnaire included 24 facets of job satisfaction. The collection of job satisfaction facets was based on the “Study of job satisfaction and trust in Slovenian police service”, which has already been used to examine satisfaction in the Slovenian police [65]. In order to simplify the analysis and add to its transparency, the items of job satisfaction were defined relatively broadly (including the highest possible number of items). The respondents had to rate the degree to which they were satisfied with specific facets of their job on a five-point scale, ranging from “extremely dissatisfied” (1) to “extremely satisfied” (5).

In the third part, the questionnaire included questions regarding interpersonal relations, leadership of the organizational unit, conflict management, commitment, and trust in leaders and an assessment of whether employees receive all information needed to perform their jobs. Each of the listed issues was covered by a specific question, e.g., regarding interpersonal relations the question was: “What are, in your opinion, the interpersonal relations?”. The possible answers were (1—very bad, 2—mainly bad, 3—indecisive, 4—mainly good, and 5—very good). This part of the survey was also taken from the “Study of job satisfaction and trust in Slovenian police service” with the aim of getting the data for longitudinal analysis. Some questions related to the police as a whole and others to a specific organizational unit (police station). The respondents had to rate their answers on a five-point scale.

Last but not least, the claims from the Common Assessment Framework (CAF) model [30,37], were used in order to obtain police service employees’ opinions on the enablers through which management should ensure the excellence of police service operations. The enablers were rated on a five-point scale, where 1 meant “in our organization we do not deal with that area, there is no care for that area, we are not active in that area” and 5 meant “in our organization, that area is excellently taken care of, all employees actively cooperate on the execution of activities in that area, that area is constantly being improved”.

Altogether, there were 90 items analyzed in the survey—10 on demographic data, 24 on facets of job satisfaction, followed by 12 questions in the third part—on relations, trust, commitment etc., 24 regarding the principles (values) of police work, and 20 on CAF enablers. The participation in the survey was anonymous and volunteer. The invitations to participate were sent to police service employees both top-down (from police managers) and bottom-up (from police service labor union). The project has had a strong support by the police managers.

### 3.2. Data Analysis

The data were analyzed with the SPSS 23.0 (IBM, New York, United States) statistical programme and its AMOS plug-in. First, descriptive statistics (average and standard deviation) and correlation analysis (Pearson coefficient correlations) were calculated in order to gather basic information about the variables. Pearson’s correlation test (r) was employed to measure the correlation between two continuous variables. Exploratory factor analysis (EFA) was used to formulate groups of job satisfaction facets, CAF enablers and elements of trust in leaders [66]. After testing for the normality of distribution, factor analysis with principal components and a varimax rotation was undertaken to examine which factors of the scale comprised coherent groups of items. The maximum likelihood method was used for the confirmatory factor analysis (CFA) in AMOS as it assumes multivariate normality [67] of the observed variables. The Kaiser criterion was applied to select the number of factors [68] and the Kaiser-Meyer-Olkin (KMO) test and Bartlett’s test of sphericity were applied to measure the sampling adequacy [69]. Structural modelling of equations (SEM), as a technique representing an extended version of many multivariate modelling techniques [70], was used as the interaction between the latent and manifest variables and their impact were studied simultaneously. The technique appears to be most appropriate in the behavioral research context, where it makes sense to consider all the characteristics that define the studied population as a complex whole rather than at the level of individual characteristics. The tendency was to first build a measurement model that would fit the data and also meet the other validity and reliability indicators, and second, to model complex relationships involving latent constructs using SEM [71].

## 4. Results

### 4.1. Descriptive Statistics and Correlation Analysis

The average values, standard deviations (descriptive statistics) and intercorrelations (correlation analysis) for the variables are presented in Table 1. There were 17 items included into the analysis, presenting 4 latent factors, and one additional observed manifest variable presenting LSOU, all measured on 1–5 scale. The results show that all pairs of variables correlate. In general, the least correlated are variables at the macro-organizational level (strategy/leadership and processes) with variables on the micro-organizational level (autonomy/tasks and trust).

### 4.2. Confirmatory Factor Analysis

After examining the exploratory factor analysis, confirmatory factor analysis was performed. The model was simplified to ensure a proper fit. In the process of elimination, variables with low factor loadings were excluded in the cases of the Autonomy/Tasks (3 out of 7 were excluded—the remaining ones are listed from A1 to A4 in Table 2) and Trust (1 out of 3 was excluded—the remaining 2 are listed as T1 and T2 in Table 2) constructs. In cases of Strategy/Leadership and Processes all variables were taken into account. Leadership style in an organizational unit could not be presented as a latent variable (factor) as there were no appropriate variables to combine with the style of the leadership in an organizational unit.

In the next step, confirmatory factor analysis was employed in order to test whether the model fits the data adequately. All factor loadings (Table 2) were higher than 0.5, indicating that all the latent variables are well represented by the indicators.

### 4.3. Reliability and Validity

In the next phase, indicators of the reliability and validity of the constructs in the model were calculated (Table 3). As explained above, not all items were included in the confirmatory factor analysis (CFA), while the exploratory factor analysis (EFA) suggested a reduction as the validity and reliability of the constructs would not be achieved if a complete set of variables were included in the questionnaire.

Composite reliability (CR) and convergent validity (AVE) were achieved in all cases. Internal consistency was identified with Cronbach’s alpha coefficient. As seen from Table 3, the factor Trust consists of a small number of items /two) so it is appropriate to consider the cut-off value of 0.6 [72] —that is also the reason why it has lower Cronbach’s alpha compared to other variables. This confirms that all the measurement scales are valid and reliable. Consequently, the questionnaire displays a high level of internal reliability to some extent. Discriminant validity is shown when each measurement item correlates weakly with all the other constructs except for the one with which it is theoretically associated [73].

### 4.4. Structural Equation Model

The fit indices show that the measurement model satisfactorily fits the data (Table 4). The absolute and incremental indices exceed the recommended values; the model is parsimonious. CMIN (with df and *p*) is not reliable given the large sample size.

The structural equation model (Figure 2) includes 4 constructs (Strategy/Leadership, Processes, Autonomy/Tasks, Trust) and 17 observed variables, which play roles as items in these factors and LSOU presented by one additional manifest variable. Strategy/Leadership and Processes are exogenous constructs, with a strong mutual correlation (0.78). Strategy/Leadership has an indirect effect on Trust with a direct effect on Autonomy/Tasks (0.58) and Leadership style in the organizational unit (LSOU) (0.22). The strongest effect is from LSOU to Trust (0.63). The model as whole explains quite a large proportion, 74%, of the variance of Trust, while the exogenous constructs explain 48% of LSOU and 34% of Autonomy/Tasks. Figure 2 presents standardized regression weights for a relative comparison of the effects’ strengths.

The unstandardized (B) regression weights (Table 5) make predictions in measurement units possible. If Autonomy/Tasks were to improve by 1 point on the 1–5 scale, this would improve Trust by only 0.16 of a point, but improving Strategy/Leadership by 1 point would affect the Autonomy/Tasks construct by about 0.75 of a point. While Strategy/Leadership indirectly affects the Trust (through LSOU and Autonomy/Tasks) it is also an important component predictor in this structural model. By improving Strategy/Leadership for 1 point, the LSOU would increase for 0.35 point, and improving Autonomy/Tasks for 1 point would increase Trust for 0.16 point. The same importance was discovered for the factor Processes because its mutual correlation with Strategy/Leadership, which later on indirectly affects Trust. It is necessary for Trust predictions that all factors are included in the model with simultaneous correlates, which is the most important added value to Structural equation modelling analytics. The results indicate that it is important to dedicate time and energy to all determinants of trust at different hierarchical levels, starting from the top down, involving managers on all levels, not only immediate superiors. Trust depends on all relations (with direct or indirect effects) between predictors/factors included in the model.

## 5. Discussion

Our results confirm the relationships between the determinants of trust at different hierarchical levels of an organization are both significant and positive. Since not many systematic research studies have been conducted on the holistic approach when studying relationships between different determinants of trust within an organization, this study provides evidence of specific linkages between these determinants. The study examined different CAF enablers, facets of job satisfaction and their impact on trust in one’s immediate leader. During our research, our hypothesized model was confirmed and in the following discussion, we put the results in perspective.

According to Mumford, Zaccaro, Fleischman and Reiter-Palmon [74], and Zaccaro [75], relatively little research has focused explicitly on leadership at the top of organizations, despite the recognized qualitative differences between upper and lower levels of organizational leadership. In our study, we first examined (H1, H2, H3) the relationship between the CAF elements, i.e., Strategy/Leadership and Processes, as assessed by the respondents. We assigned them to the macro-organizational level. A strong mutual correlation between the two enablers was discovered, implying it is important to focus on different enablers in business excellence models (e.g., CAF). This confirms the findings of Aladwan and Forrester [33], Arau’jo and Sampaio [76], Suárez et al. [32] and Vakalopoulou et al. [30] which state that leadership, policy and strategy influence the other enablers of business excellence models. Second (H5), the results confirm those of prior studies regarding the significance of the leadership style in an organizational unit (in our study, this factor represented the mezzo-organizational level, i.e., the leadership of a police station) when studying the determinants of trust, thus supporting the findings of Dirks [41], Lee et al. [5] and Vito and Higgins [77]. Third, we found that satisfaction with the autonomy of tasks (deriving from the micro-organizational level (level of jobs) influences the feeling of trust directly and significantly (H6) and indirectly through the leadership style in an organizational unit (H4 and H5)—the mezzo-organizational level (middle management).

In the last step, the study also looked at potential differences in the respondents’ demographic characteristics. Despite finding some differences for all multiple groups (gender—male/female, working in the office or in the field, years of service in the police—less than 10 years or more than 10 years, distance from home to work—less than 50 km or more than 50 km), the general model remains valid. The differences were not so big that they would make the model no longer fit. In sum, this study suggests the relationship between the elements of management/leadership at different organizational levels and trust is complex, and that simple strategies and approaches focusing on a single organizational level may be ineffective.

### Limitations and Future Research

While our study brings important findings, it does have some limits. First, while the sample was numerous (*n* = 1209), it was not statistically representative of the population (*n* = 8808). This is probably due to the fear of anonymity because the survey was performed online. Nevertheless, the response rate was above what was expected and quite high compared to the samples of other studies, which was a result of the bottom-up (by the trade-unions) and top-down (by the management) communication to and motivation for police service employees to join the survey. Second, the study was limited to the Slovenian police service and may not fully represent other countries’ police organizations. Therefore, the findings should be understood with some degree of caution in terms of their generalizability. Third, the survey was launched after the economic crisis which had even deteriorated the material/financial situation of the police as an institution (in terms of available funding, equipment—cars, uniforms, premises, etc.) and of the police service employees themselves (salaries and promotion). It may be assumed the employees were strongly influenced by that situation when completing the questionnaire, which might mean different survey findings would be obtained if the survey were to be performed a few years later. Statistically, LSOU should be reconsidered and tested as a latent variable/construct in future research to improve the model.

Further research on the areas covered in our research could concentrate on an international comparison of police organizations, on comparison of police with other public administration organizations and longitudinal analysis in order to discover how the determinants of trust change over time. Further investigation may also need to expand the range of elements of trust at each organizational level and delve into the ways the positive results concerning employees’ trust in their managers can improve the results of the police service as a whole. Analyzing the opposite—the trust managers have in their employees—would also be an interesting topic for empirical inquiry.

## 6. Conclusions

The findings of this study present key implications for both researchers and members of the police service work environment. While budgetary limitations often serve as an excuse for slow expansion and development, much can be done without larger investments. Police managers are challenged to develop and expand managerial approaches at different organizational levels in order to increase trust in superiors. Training programmes on use of the CAF and on leadership approaches and organization should be offered in certain periods to specific target groups. On that basis, the CAF should be understood and systematically used as a self-evaluation and self-improvement tool at all levels of the police hierarchy. Leadership trainings should be tailor-made for specific groups of existing leaders and for those set to become leaders in the following years (talent management, succession planning). Training on organization should, similarly as that on the CAF, be offered to all employees and adapted to the needs of each hierarchical level. Of course, the effect of all of those trainings should be measured and the training approaches should be further refined according to the results of those measurements. Schafer [78] claims that, if money were the key limitation, external initiatives could overcome that barrier. Culture, politics, structure and inertia might be more difficult to overcome. That is why systematic, holistic and sustainable training programmes should be created for all police service employees.

While several studies have already been performed, much time and effort still need to be invested in research in the fields of management, organization, excellence models, job satisfaction and leadership in policing. Police service employees are a specific and very important group of public administration employees and it is therefore important to understand the determinants and consequences of the many phenomena that are specific for their work environment.

Despite the above limitations, the study results provide an important insight into the determinants of trust at different organizational levels. They confirm the need to investigate the determinants of trust at different organizational levels because the police service is a complex hierarchical system in which each hierarchical level considerably influences a specific phenomenon in a different way. In our study, we identified a unique combination for determining trust that provides important insights regarding how employees perceive the elements of trust from various organizational levels.

Due to a limited number of predictors of trust on each organizational level, there is a need to further investigate trust. It is important that both researchers and police employees understand that it is crucial to strengthen/improve the elements of trust at different organizational levels, not just in the direct relationship between the manager and the employee.

The results of our study suggest police management should systematically and holistically develop and implement management approaches at all organizational levels—especially focusing on the strategy, processes, leadership, and organization of work. The results therefore enable a better understanding of employees’ perceptions of specific management approaches, in turn opening up an opportunity for more focused research, measurement, and implementation.

## Figures and Tables

**Figure 1 ijerph-16-02525-f001:**
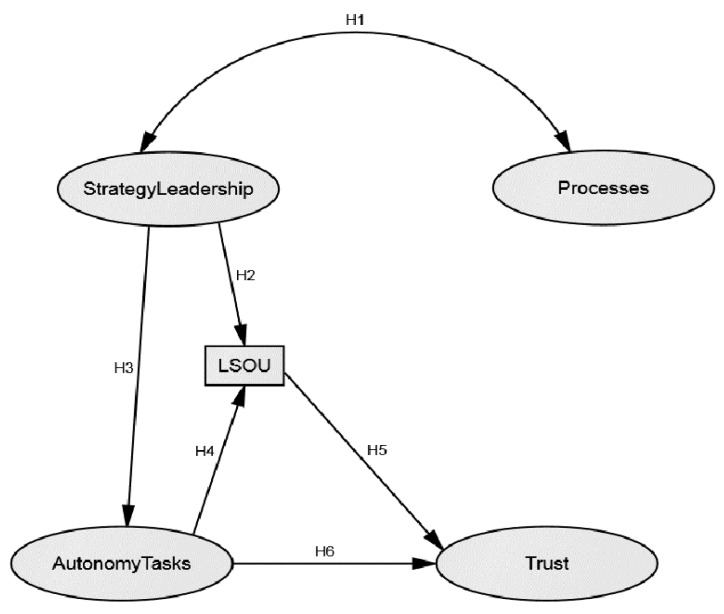
Hypothesized model of the determinants of trust in leaders.

**Figure 2 ijerph-16-02525-f002:**
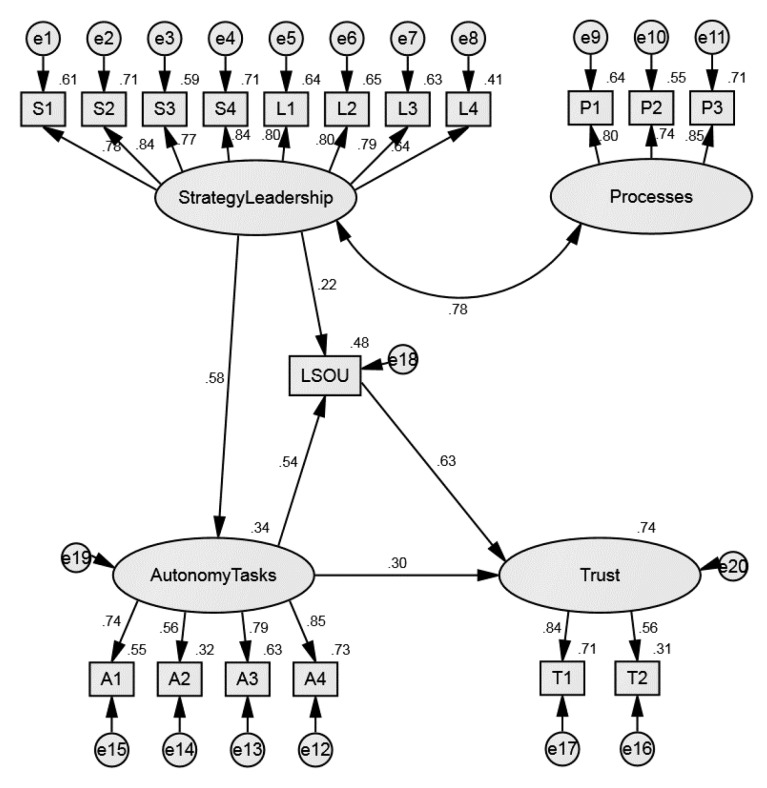
Structural equation model.

**Table 1 ijerph-16-02525-t001:** Average values (M), standard deviations (SD) and correlations between variables.

Constructs and Variables	M	SD	1	2	3	4	5	6	7	8	9	10	11	12	13	14	15	16	17	18
*Strategy/Leadership*	2.80	1.09																		
(S1) stakeholder analysis	2.8	1.00	1																	
(S2) strategy design	2.73	1.02	0.749 **	1																
(S3) strategy implementation	2.71	1.06	0.582 **	0.671 **	1															
(S4) planning, implementing and monitoring innovation	2.55	1.07	0.686 **	0.752 **	0.747 **	1														
(L1) mission, vision, values	2.97	1.06	0.579 **	0.641 **	0.588 **	0.619 **	1													
(L2) management, change management	2.72	1.10	0.588 **	0.658 **	0.580 **	0.639 **	0.750 **	1												
(L3) motivation, support, role model	2.88	1.24	0.580 **	0.609 **	0.566 **	0.639 **	0.684 **	0.690 **	1											
(L4) stakeholder relationships	3.05	1.13	0.544 **	0.538 **	0.464 **	0.499 **	0.517 **	0.497 **	0.574 **	1										
*Processes*	2.73	1.04																		
(P1) process design, implementation and improvement	2.53	1.1	0.535 **	0.587 **	0.549 **	0.622 **	0.496 **	0.502 **	0.525 **	0.402 **	1									
(P2) design of customer-focused products and services	2.93	0.99	0.503 **	0.466 **	0.475 **	0.462 **	0.402 **	0.403 **	0.398 **	0.468 **	0.557 **	1								
(P3) process innovation	2.74	1.02	0.556 **	0.558 **	0.500 **	0.548 **	0.483 **	0.495 **	0.467 **	0.458 **	0.667 **	0.662 **	1							
*Autonomy/Tasks*	3.13	1.09																		
(A1) possibility of performing work autonomously	3.02	1.05	0.311 **	0.351 **	0.333 **	0.337 **	0.403 **	0.381 **	0.369 **	0.305 **	0.291 **	0.306 **	0.308 **	1						
(A2) variety of tasks	3.54	1.04	0.210 **	0.222 **	0.214 **	0.232 **	0.308 **	0.278 **	0.261 **	0.243 **	0.176 **	0.220 **	0.206 **	0.488 **	1					
(A3) possibility of realizing one’s abilities	3.05	1.08	0.318 **	0.350 **	0.317 **	0.355 **	0.437 **	0.405 **	0.443 **	0.329 **	0.311 **	0.259 **	0.300 **	0.586 **	0.452 **	1				
(A4) possibility of participating in decision-making on organization2	2.92	1.18	0.319 **	0.384 **	0.351 **	0.366 **	0.478 **	0.452 **	0.485 **	0.346 **	0.317 **	0.272 **	0.304 **	0.619 **	0.460 **	0.687 **	1			
*(LSOU) Leadership style in organizational unit*	3.28	1.25	0.335 **	0.399 **	0.364 **	0.387 **	0.531 **	0.478 **	0.566 **	0.345 **	0.317 **	0.278 **	0.292 **	0.471 **	0.365 **	0.519 **	0.594 **	1		
*Trust*	3.65	1.00																		
(T1) received necessary information	3.66	1.06	0.317 **	0.381 **	0.336 **	0.376 **	0.501 **	0.447 **	0.527 **	0.309 **	0.297 **	0.239 **	0.268 **	0.429 **	0.284 **	0.438 **	0.523 **	0.704 **	1	
(T2) trusting immediate leader	3.64	0.95	0.298 **	0.355 **	0.330 **	0.352 **	0.387 **	0.365 **	0.391 **	0.320 **	0.259 **	0.216 **	0.271 **	0.386 **	0.222 **	0.367 **	0.373 **	0.439 **	0.470 **	1

Note: ** *p* < 0.01.

**Table 2 ijerph-16-02525-t002:** Standardized factor loadings.

Variable		Construct	λ
(S1) stakeholder analysis	←	Strategy/Leadership	0.780
(S2) strategy design	←		0.842
(S3) strategy implementation	←		0.768
(S4) planning, implementing and monitoring innovation	←		0.839
(L1) mission, vision, values	←		0.801
(L2) management, change management	←		0.804
(L3) motivation, support, role model	←		0.793
(L4) stakeholder relationships	←		0.641
(P1) process design, implementation and improvement	←	Processes	0.798
(P2) design of customer-focused products and services	←		0.739
(P3) process innovation	←		0.846
(A1) possibility of performing work autonomously	←	Autonomy/Tasks	0.741
(A2) variety of tasks	←		0.566
(A3) possibility of realizing one’s abilities	←		0.796
(A4) possibility of participating in decision-making on organization	←		0.855
(T1) received necessary information	←	Trust	0.829
(T2) trusting immediate leader	←		0.565

**Table 3 ijerph-16-02525-t003:** Validity and reliability indicators.

Variable		Construct	Composite Reliability (CR)	Convergent Validity (AVE)	Cronbach’ α
(S1) stakeholder analysis	←	Strategy/Leadership	0.958	0.618	0.928
(S2) strategy design	←				
(S3) strategy implementation	←				
(S4) planning, implementing and monitoring innovation	←				
(L1) mission, vision, values	←				
(L2) management, change management	←				
(L3) motivation, support, role model	←				
(L4) stakeholder relationships	←				
(P1) process design, implementation and improvement	←	Processes	0.835	0.902	0.835
(P2) design of customer-focused products and services	←				
(P3) process innovation	←				
(A1) possibility of performing work autonomously	←	Autonomy/Tasks	0.893	0.557	0.830
(A2) variety of tasks	←				
(A3) possibility of realizing one’s abilities	←				
(A4) possibility of participating in decision-making on organization	←				
(T1) received necessary information	←	Trust	0.637	0.765	0.637
(T2) trusting immediate leader	←				

**Table 4 ijerph-16-02525-t004:** Model fit.

Indicator	Abbreviation	Recommended Value	Value
Minimum of Discrepancy (χ^2^)	CMIN		1040
Degrees of Freedom	df		130
*p* value	*p*	>0.05	0
Goodness of Fit Index	GFI	>0.90	
Adjusted Goodness of Fit Index	AGFI	>0.90	
Root Mean Square Error of Approximation	RMSEA	<0.05 or 0.08	0.076
Bentler-Bonett Index/Normed Fit Index	NFI	>0.90	0.918
Comparative Fit Index	CFI	>0.90	0.927
Tucker Lewis Index/Non-Normed Fit Index	TLI or NNFI	>0.90	0.904
Parsimonious Normed Fit Index	PNFI	>0.60	0.698

**Table 5 ijerph-16-02525-t005:** Regression weights and predictions.

% Variance Explained	Unstandardized and Standardized Regression Weights			B	β
34%	Autonomy/Tasks	<	Strategy/Leadership	0.754	0.584
48%	LSOU	<	Strategy/Leadership	0.346	0.217
		<	Autonomy/Tasks	0.669	0.542
74%	Trust	<	LSOU	0.265	0.626
		<	Autonomy/Tasks	0.158	0.302

Note: LSOU—Leadership Style in Organizational Unit.

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
