# Peer review of "Factors of Trust in Immediate Leaders: An Empirical Study in Police Service Environment"

_ijerph, 2019, doi:10.3390/ijerph16142525_

Round 1
Reviewer 1 Report
The abstract promises a more cohesive piece of research than that presented. I suspect the challenge has been in the writing; nevertheless, it is difficult to commend the paper for publication in its current state. As I briefly outline below, there are challenges with clarity in defining and conceptualising the main variable trust – and critically in understanding how it is measured here, and then analysed. That is, the paper needs some work on its structure, level of explanation, and critically, sufficient detail to be replicable and penetrate the priorities of any recommendations that come out of this work.
Introduction: Succinct and focused background to this study, but could be much clearer.
The rationale for this research could be stronger if key point (lines 56-57) was not the sweeping assertion “Trust is often extremely difficult to achieve and maintain because people by nature do not trust each other”. I suspect this may be the authors’ own view – which is why there is no supporting reference – but I suggest this can be strongly contested. Indeed, I can point to literature in a variety of settings that suggests otherwise.
There is a fundamental issue that trust is not adequately defined as a multidimensional construct. We are told how it has been examined previously (pp59-61) – but again, without references to uphold assertions. It is true that there are many definitions of trust (line 76) in the literature, and whilst the authors do portray trust in this way, it is quite difficult to penetrate – especially when we read that trust encompasses five important parts, and two of these are trust and self-trust (lines 78-80). Ultimately there are assertions that trust is an emotion (line 76), an expression (line 86), a state (line 88), and in contrast to their point I mention above, “People are initially trusting of other people …” (line 93). A clearer outline of how (and why) trust is conceptualised in this research is required.
It would be beneficial to define “facets of job satisfaction” (p127) rather than leave the reader to assume what these are.
Grammar issues – especially tense errors in lines 56-66 which compromises argument; syntax and punctuation, eg pp102-104; 110;
Lines 171-2 – organisation of sentence
Figure 1 (lines 198-200) needs to be bigger.
Referencing weak – various assertions without references, and style not strictly Vancouver.
Methods
Critical challenge is the lack of replicability from information given.
§ It is very difficult to penetrate how key dependent variable (Trust) is measured. This is in a part of “in the third part of the questionnaire (lines 223-227) without any clarification of how many items, example items, nature of five point scale, where items were derived, (and why), etc. The same can be said for all aspects of the survey in the third part.
§ Should provide total number of items altogether on the survey. It is not even clear how many items are on the previously validated parts of the survey. Critically, I cannot use reference given for Job Satisfaction Scale in so far as in Slovenian, and this is why more detail is required, and there is no reference for the questions from the CAF (line 228).
§ Whilst sample size is given, response rate is not. I know such surveys generally have a low response rate, but it is important to report it here – rather than as a limitation in the discussion.
Line 208 “Their work mainly (28.5%) entailed…..” I suggest the main part is unexplained; this is not even one-third.
Results
I suggest structure of this section would be improved for illustrating measures were reliable before inferential correlation analysis.
Line 257 “All pairs of variables correlate”. Correlation matrix implies two aspects of trust measured; in view of focus on these, some comment would have been useful. In particular, very high relationship of LSOU and T1 (.704). This then is biggest predictor of Trust when other factors are included in the model. In light of this, the summary at lines 303-305 could be a better reflection of given results, if one were to prioritise efforts to improve trust.
Discussion
Succinct.
The Conclusion seems to be an adjunct to the research – especially the recommendations concerning training. The point (line 373-376) that “everybody is exposed to … some kind of stress”, and “Stress management should be introduced” arises rather baldly, and the fit with trust not particularly explicit. (I also note that if pressure extended to stress in the UK, there would be a legal duty of care to intervene, which clearly officers in Slovenia cannot rely on.)
Various undefined acronyms (CEO, HRM, EFQM, LSOU)
Author Response
The abstract promises a more cohesive piece of research than that presented. I suspect the challenge has been in the writing; nevertheless, it is difficult to commend the paper for publication in its current state. As I briefly outline below, there are challenges with clarity in defining and conceptualising the main variable trust – and critically in understanding how it is measured here, and then analysed. That is, the paper needs some work on its structure, level of explanation, and critically, sufficient detail to be replicable and penetrate the priorities of any recommendations that come out of this work.
Introduction: Succinct and focused background to this study, but could be much clearer.
The rationale for this research could be stronger if key point (lines 56-57) was not the sweeping assertion “Trust is often extremely difficult to achieve and maintain because people by nature do not trust each other”. I suspect this may be the authors’ own view – which is why there is no supporting reference – but I suggest this can be strongly contested. Indeed, I can point to literature in a variety of settings that suggests otherwise.
Taken into account – restructured.
There is a fundamental issue that trust is not adequately defined as a multidimensional construct. We are told how it has been examined previously (pp59-61) – but again, without references to uphold assertions. It is true that there are many definitions of trust (line 76) in the literature, and whilst the authors do portray trust in this way, it is quite difficult to penetrate – especially when we read that trust encompasses five important parts, and two of these are trust and self-trust (lines 78-80). Ultimately there are assertions that trust is an emotion (line 76), an expression (line 86), a state (line 88), and in contrast to their point I mention above, “People are initially trusting of other people …” (line 93). A clearer outline of how (and why) trust is conceptualised in this research is required.
Taken into account – restructured.
It would be beneficial to define “facets of job satisfaction” (p127) rather than leave the reader to assume what these are.
Taken into account – explained.
Grammar issues – especially tense errors in lines 56-66 which compromises argument; syntax and punctuation, eg pp102-104; 110;
Taken into account – corrected/amended. Thank you very much!
Lines 171-2 – organisation of sentence
Taken into account – corrected.
Figure 1 (lines 198-200) needs to be bigger.
Taken into account – enlarged.
Referencing weak – various assertions without references, and style not strictly Vancouver.
Taken into account – some assertions deleted, referencing style corrected.
Methods
Critical challenge is the lack of replicability from information given.
§ It is very difficult to penetrate how key dependent variable (Trust) is measured. This is in a part of “in the third part of the questionnaire (lines 223-227) without any clarification of how many items, example items, nature of five point scale, where items were derived, (and why), etc. The same can be said for all aspects of the survey in the third part.
Taken into account – added in 3.1.
§ Should provide total number of items altogether on the survey. It is not even clear how many items are on the previously validated parts of the survey. Critically, I cannot use reference given for Job Satisfaction Scale in so far as in Slovenian, and this is why more detail is required, and there is no reference for the questions from the CAF (line 228).
Taken into account – added in 3.1.
§ Whilst sample size is given, response rate is not. I know such surveys generally have a low response rate, but it is important to report it here – rather than as a limitation in the discussion.
Taken into account – added in 3.1.
Line 208 “Their work mainly (28.5%) entailed…..” I suggest the main part is unexplained; this is not even one-third.
Taken into account – added in 3.1.
Results
I suggest structure of this section would be improved for illustrating measures were reliable before inferential correlation analysis.
Line 257 “All pairs of variables correlate”. Correlation matrix implies two aspects of trust measured; in view of focus on these, some comment would have been useful. In particular, very high relationship of LSOU and T1 (.704). This then is biggest predictor of Trust when other factors are included in the model. In light of this, the summary at lines 303-305 could be a better reflection of given results, if one were to prioritise efforts to improve trust.
Taken into account – added in 4.3.
Discussion
Succinct.
The Conclusion seems to be an adjunct to the research – especially the recommendations concerning training. The point (line 373-376) that “everybody is exposed to … some kind of stress”, and “Stress management should be introduced” arises rather baldly, and the fit with trust not particularly explicit. (I also note that if pressure extended to stress in the UK, there would be a legal duty of care to intervene, which clearly officers in Slovenia cannot rely on.)0
Taken into account – deleted.
Various undefined acronyms (CEO, HRM, EFQM, LSOU)
Taken into account – explained.
Reviewer 2 Report
The proposed model is interesting. However, it is strange that the process variable does not relate either directly or indirectly to trust. Then, why should it be considered a determinant of trust in your model? Could you justify its relevance?
Author Response
The proposed model is interesting. However, it is strange that the 'process' variable does not relate either directly or indirectly to trust. Then, why should it be considered a determinant of trust in your model? Could you justify its relevance?
There are 5 sets of Approaches in the Common Assessment Framework model (+ 4 sets of Results). These are (1) Leadership, (2) Strategy and planning, (3) People, (4) Partnerships and resources, and (5) Processes. We found Strategy, together with Leadership, and Processes as those sets of Approaches that are designed and determined on the top of the hierarchical pyramid (approaches within Processes: (1) continuous detection, design, management and improvement of processes, (2) development and assurance of services directed towards the citizens, (3) innovation of the processes into which the citizens are involved etc.). The measures that are taken i to account within Processes are of big importance for efficient functioning of an organization.
Because there was a mutual correlation between Strategy/Leadership and Processes (see 4.3) and consequently Strategy’s/Leadership’s indirect influence on LSOU and on Autonomy/Tasks, and then on Trust, we found it important to include Processes into the model.
Basic idea of structural equation modelling is to investigate simultaneous impacts of all into the model included variables - both items that represent factors and factors which are then interrelated. If any of the variables in the model is changed or excluded, all parameters in the model are changed. Processes didn’t have a direct impact – but with a positive correlation with S/L they indirectly influence Trust. A share of explanatory variance is high (70 %), which means that such a model has a high explanatory power and value. If the Processes were excluded, the strength of the model would have dropped.
Reviewer 3 Report
I realize that a great work and time has been devoted to this paper. This is a topic of great significance to emotional wellbeing of polices that can affect the quality of life, and therefore, be a public health problem. So I appreciate authors examining this topic.
The paper has a lot of strengths but I think that some changes should be recommended.
Abstract:
Readers should be able to read the abstract in isolation and understand what you have done, and its implications. So I recommend to the authors to expand the “Results” with more information, because it is unclear which correlations are significant.
I would suggest to the authors to avoid the word “we/our”, writing in impersonal mode as scientific style.
Please, avoid using abbreviations in the abstract and in the keywords.
I would suggest to the authors to check the keywords, which would be indexed in MeSH or DeCS terms.
Introduction:
Please, write “study” instead of “paper” in abstract and in the introduction.
The introduction is very dense, very long, disorganized, repetitive and does not justify why it has conducted this study.
I would suggest to the authors to avoid the word “we/our”, writing in impersonal mode as scientific style.
The first paragraph of the introduction has not any reference.
There are some paragraphs that would be deleted, because repeats the same that it is in the abstract or in the method and it is unclear and poorly written (like paragraph between line 67-73).
In addition to this, the aim of study would be placed at the end of introduction, not in the middle.
In line 52-53, please specify which year is the meta-analysis that you talk about.
I would suggest to the authors that do not put abbreviations which has not been described before, as “HRM” in line 115 that I suppose is Human Resources Management, cited in line 39. The same occurs with “LSOU”, so the reader does not know the meaning.
The figure 1 is too small and poor quality.
Methodology:
It is unclear the design of study.
The authors must specify if the participation was anonymous and volunteer and if there is consent of Ethics Committee and if the participants had to sign an informed consent.
I suggest to the authors to specify the response rate in percentages.
It is unclear if the instruments used were ad-hoc. Please, you have to specify this in the text.
I don’t know if CAF is a model or a questionnaire. I suggest to the authors to specify this in the text and please, explain better.
If CAF is a questionnaire, the author must have to write the validation to Slovenian population and the Cronbach’s Alpha Coefficient.
Results, Discussion and Conclusions:
The tables are not well described in the text and they not have footnotes for the abbreviations.
Could you explain why the Cronbach’s Alpha Coefficient of Trust’s dimension is so low?
The table 5 does not meet standards of tables’ format. So please, check it.
References:
References do not meet the standards of the Journal, so please consult the following link:
https://www.mdpi.com/journal/ijerph/instructions
I hope that these recommendations do not discourage the authors and I want to recommend the authors to continue working on this paper.
Author Response
I realize that a great work and time has been devoted to this paper. This is a topic of great significance to emotional wellbeing of polices that can affect the quality of life, and therefore, be a public health problem. So I appreciate authors examining this topic.
The paper has a lot of strengths but I think that some changes should be recommended.
Abstract:
Readers should be able to read the abstract in isolation and understand what you have done, and its implications. So I recommend to the authors to expand the “Results” with more information, because it is unclear which correlations are significant.
Taken into account – reformulated.
I would suggest to the authors to avoid the word “we/our”, writing in impersonal mode as scientific style.
Taken into account – corrected.
Please, avoid using abbreviations in the abstract and in the keywords.
Taken into account – corrected.
I would suggest to the authors to check the keywords, which would be indexed in MeSH or DeCS terms.
Taken into account – reorganized.
Introduction:
Please, write “study” instead of “paper” in abstract and in the introduction.
Taken into account – corrected.
The introduction is very dense, very long, disorganized, repetitive and does not justify why it has conducted this study.
Taken into account –shortened.
I would suggest to the authors to avoid the word “we/our”, writing in impersonal mode as scientific style.
Taken into account – corrected.
The first paragraph of the introduction has not any reference.
It is authors’ observation.
There are some paragraphs that would be deleted, because repeats the same that it is in the abstract or in the method and it is unclear and poorly written (like paragraph between line 67-73).
The structure of the paper, described in the Introduction, is usually an essential part of Introduction. Some detailed explanations were deleted in order to make it more smooth J.. Thank you!
In addition to this, the aim of study would be placed at the end of introduction, not in the middle.
The aim is placed just before the description of the paper’s structure.
In line 52-53, please specify which year is the meta-analysis that you talk about.
Taken into account – added.
I would suggest to the authors that do not put abbreviations which has not been described before, as “HRM” in line 115 that I suppose is Human Resources Management, cited in line 39. The same occurs with “LSOU”, so the reader does not know the meaning.
Taken into account – corrected.
The figure 1 is too small and poor quality.
Taken into account – corrected.
Methodology:
It is unclear the design of study.
Taken into account – added in 3.1 and 4.1.
The authors must specify if the participation was anonymous and volunteer and if there is consent of Ethics Committee and if the participants had to sign an informed consent.
Taken into account – added in 3.1.
I suggest to the authors to specify the response rate in percentages.
Taken into account – added in 3.1.
It is unclear if the instruments used were ad-hoc. Please, you have to specify this in the text.
Taken into account – added in 3.1.
I don’t know if CAF is a model or a questionnaire. I suggest to the authors to specify this in the text and please, explain better.
If CAF is a questionnaire, the author must have to write the validation to Slovenian population and the Cronbach’s Alpha Coefficient.
CAF is not a questionnaire but a self-evaluation business excellence model (designed for public administration), similar as EFQM in Europe or MBNQA in the USA for all types of organizations.
Results, Discussion and Conclusions:
The tables are not well described in the text and they not have footnotes for the abbreviations.
Taken into account – Table 2 described additionally.
It is not clear which abbreviations are not explained – if not in the table, they are explained in the text.
Could you explain why the Cronbach’s Alpha Coefficient of Trust’s dimension is so low?
Taken into account – additionally explained in 4.3.
The table 5 does not meet standards of tables’ format. So please, check it.
Taken into account – corrected. Thank you!
References:
References do not meet the standards of the Journal, so please consult the following link:
https://www.mdpi.com/journal/ijerph/instructions
Taken into account – referencing style corrected.
I hope that these recommendations do not discourage the authors and I want to recommend the authors to continue working on this paper.
Many thanks for your valuable corrections/suggestions.
Round 2
Reviewer 1 Report
I am pleased that the authors were able to appropriately respond to my original comments: I am generally satisfied that the amendments address the points of clarity and precision I raised.
Two small points I suggest would better if tweaked are:
Line 124 - why not simply list all facets of Job Satisfaction measured rather than etc? (Never assume the reader will know what these are in this context).
Lines 246-7 - yes, report that the project had strong support by the police service managers (edited grammar), but I don't really think you can consider that this resulted in a high number of responses when the response rate was 13.7% (line 203). The sample size was good enough by dint of being a large organisation, not especially from support of managers, it seems to me. That is, I suggest deletion of "which resulted in a high number of responses."
There are new grammatical and spelling errors:
Line 328 (change a to the)
Line 330 modelling
Line 331 analytics
Line 333 I don't think "between" is the right word - being?
Author Response
RESPONSES TO THE REVIEWERS' COMMENTS - 2
Dear Reviewer,
Many thanks for additional recommendations!
Kind regards,
Authors
REVIEWER 1
I am pleased that the authors were able to appropriately respond to my original comments: I am generally satisfied that the amendments address the points of clarity and precision I raised.
Two small points I suggest would better if tweaked are:
Line 124 - why not simply list all facets of Job Satisfaction measured rather than etc? (Never assume the reader will know what these are in this context).
Thank you, taken into account – all 24 facets are listed. There were 24 of them and we didn’t list them because we thought there would be too much text J.
Lines 246-7 - yes, report that the project had strong support by the police service managers (edited grammar), but I don't really think you can consider that this resulted in a high number of responses when the response rate was 13.7% (line 203). The sample size was good enough by dint of being a large organisation, not especially from support of managers, it seems to me. That is, I suggest deletion of "which resulted in a high number of responses."
Thank you, taken into account – deleted.
There are new grammatical and spelling errors:
Line 328 (change a to the)
Thank you, taken into account – corrected.
Line 330 modelling
Thank you, taken into account – corrected.
Line 331 analytics
Thank you, taken into account – corrected.
Line 333 I don't think "between" is the right word - being?
Thank you, taken into account – corrected.
Reviewer 3 Report
The authors have successfully fulfilled all recommendations and suggestions. Even so, the authors must specify in the text the type of design in material and methods.
Congratulations for the work.
Author Response
Dear reviewer,
thank you for the comment, we added the type of design/research in chapter 3.1. (the first sentence)
Yours Sincerely,
Authors